# Selected miRNA and Psoriasis—Cardiovascular Disease (CVD)—Overweight/Obesity Network—A Pilot Study

**DOI:** 10.3390/ijms241813916

**Published:** 2023-09-10

**Authors:** Anna Michalak-Stoma, Katarzyna Walczak, Michał Adamczyk, Małgorzata Kowal, Dorota Krasowska

**Affiliations:** Chair and Department of Dermatology, Venereology and Pediatric Dermatology, Medical University of Lublin, ul. Staszica 16, 20-081 Lublin, Poland; katarzyna.walczak@umlub.pl (K.W.); michaladamczyk1310@wp.pl (M.A.); kowalma71@o2.pl (M.K.); dorota.krasowska@umlub.pl (D.K.)

**Keywords:** psoriasis, psoriatic arthritis (PsA), overweight, obesity, cardiovascular disease (CVD), miR-22-3p, miR-133a-3p, miR-146a-5p, miR-369-3p, Let-7b-5p

## Abstract

Psoriasis is nowadays recognized as a multifactorial systemic disease with complex and not fully understood pathogenesis. In psoriatic patients, the increased cardiovascular disease (CVD) risk and frequent comorbidities like obesity are observed. The aim of this study was to investigate differences in miRNA (miR-22-3p, miR-133a-3p, miR-146a-5p, miR-369-3p, and Let-7b-5p) involved in CVD risk among psoriatic patients with overweight/obesity and with normal weight. The study comprised 28 male psoriatic patients and 16 male healthy controls. miRNA isolated from peripheral blood mononuclear cells was reverse-transcribed and RT-qPCR was performed. We have found decreased levels of miR-22, miR-133a, miR-146a, and miR-369 among the psoriatic patients. There was a statistically significant difference in miR-22 and miR-146a levels between psoriatic patients with overweight/obesity and with normal weight. There were positive correlations between miR-22 and miR-146a levels and psoriatic arthritis (PsA) in psoriatic patients with normal weight and between the miR-133a level and PsA in the overweight/obese patients. The decreased levels of selected miRNA are consistent with the levels observed in CVD indicating their impact on the CVD risk in psoriatic patients. miR-22 and miR-146 may be recognized as one of the contributing factors in the obesity-CVD-psoriasis network.

## 1. Introduction

Psoriasis is nowadays recognized as a multifactorial systemic disease with complex and still not fully understood pathogenesis. In psoriatic patients, one observes the increased cardiovascular risk and frequent comorbidities such as obesity, metabolic syndrome (MS), non-alcoholic fatty liver disease, type 2 diabetes mellitus (DM-2), inflammatory bowel disease, and depression [1,2]. This is an escalating problem in this population and the aim of many studies is to find out the complicated immune, genetic, and environmental connections.

The knowledge about the immunopathogenesis and genetics of psoriasis has developed over time. In particular, the exploration of the key role of interleukin 23 (IL-23)/T helper (Th) 17 axis opened up new possibilities for immune-based targeted therapy for psoriasis [2,3]. Multiple gene loci responsible for psoriasis susceptibility have been described as psoriasis susceptibility locus 1–15 (PSORS1–PSORS15) [2,4,5]. The *HLA-Cw*06:02* allele located in the PSORS1 locus within the major histocompatibility complex (MHC) region on chromosome 6p21.3 has been indicated to have the strongest association with psoriasis development [6,7]. Some of the psoriasis susceptibility loci like PSORS2, PSORS3, and PSORS4 have been reported to be related to genetic predisposition to MS, DM-2, family hyperlipidemia, and cardiovascular disease (CVD). Obesity seems to be crucial for the development of psoriasis comorbidities [8]. Genome-wide association studies (GWAS) in psoriatic patients have identified many genes located in non-coding DNA, involved in skin barrier function, IL-23/Th17 signaling, interferon (IFN), and nuclear factor-kappa B (NF-κB) signaling along with innate immunity, adaptive immunity, and Th-2 activation response [4,5,6,7,9].

Furthermore, not only genetic predisposition but also many environmental factors like common infections, stress, gut dysbiosis, some medications, alcohol consumption, smoking, and obesity can be responsible for the onset of psoriasis in predisposed individuals and can modulate the disease course and also enhance the cardiovascular risk [1,9,10].

The increasing number of studies highlight the impact of epigenetic factors in psoriasis [1,5,7,11,12]. Epigenetic factors can regulate gene expression at the transcriptional (via DNA methylation) and post-transcriptional levels (via microRNAs–miRNAs and long non-coding RNAs–lncRNA) [1,13]. miRNAs are single-stranded, small non–protein-coding RNAs, which regulate gene expression by binding to the 3′ UTR of mRNA, forming miRNA-mRNA complex and leading to degradation of mRNA [14]. miRNAs can regulate various cellular processes by influencing epigenetic modifications [13,14]. Over 250 miRNAs have been described to have an impact on the pathogenesis of psoriasis [2,15,16,17,18,19,20,21,22]; however, the connections between psoriasis and comorbidities are still not well recognized. 

There are observations that overweight and obesity can be associated with psoriasis; furthermore, it can be a risk factor for psoriasis onset and frequent worse response to treatment among psoriatic patients [23,24,25,26,27,28,29,30,31]. The adipose tissue is the biggest endocrine organ producing proinflammatory cytokines (tumor necrosis factor α (TNF-α), IL-6, IL-17) and adipokines which are involved in dyslipidemia, insulin resistance (IR), diabetes, and feather CVD development [26,32]. There are also T cells in the adipose tissue recognized which stimulate inflammatory response [26,33,34]. Increased production of IL-6 by adipose tissue macrophages promotes an expansion of the Th17 population [34]. According to many studies, Th17 and IL-17 are the important link between inflammation, autoimmune response, and obesity [34]. IL-6 and IL-17A regulate adipocyte differentiation and their capability to the synthesis of adipokines and chemokines [35]. An increasing number of miRNAs have been identified to play key regulatory roles in adipose tissue and, consequently, they can be potential therapeutic targets for obesity and obesity-related cardiometabolic diseases [36,37,38].

The cardiometabolic risk in people with obesity is often observed in patients with severe psoriasis too [39]. It can result from the fact that some proinflammatory cytokines are common for both psoriasis and obesity. Inflammation associated with psoriasis is enhanced by proinflammatory cytokines, and adipokines produced by adipose tissue induce IR, oxidative stress, and endothelial cell damage. Endothelial impairment increases the risk of atherosclerosis and faster development of CVD [7,40,41]. 

The aim of this study was to investigate the differences in miRNAs involved in CVD risk among psoriatic patients with particular emphasis on overweight and obese individuals. We have selected miRNA involved in the CVD (miR-22-3p, miR-133a-3p, miR-146a-5p, miR-369-3p, and Let-7b-5p), and assessed their levels in two groups of patients with psoriasis with obesity or overweight and with normal weight, as well as in the healthy control group.

## 2. Results

In the constructed miRNA-disease network, the associations between miR-22, miR-133a, miR-146a, Let-7b, and cardiovascular disease were recognized, as well as miR-22, miR-146a, miR-369, and psoriasis, and miR-146, Let-7b, and obesity (Figure 1). For the clarity of the figure, the remaining nodes have not been marked. A detailed description of all nodes is added as Appendix A.

Groups of psoriatic patients were divided based on BMI and WHR. In the psoriatic patients Group 1 BMI was higher (31.3 ± 3.9) than in Group 2 (23.8 ± 1.0) and the control group (24.0 ± 2.4). However, no statistically significant difference between BMI in Group 2 and in the control group was observed (Figure 2a). There was also a statistically significant difference for WHR between Group 1 and Group 2, and the control group, with no statistically significant difference between Group 2 and the control group (Figure 2b). There were no statistically significant differences in BMI and WHR in Group 1 and Group 2 for patients with psoriatic arthritis (PsA). In both psoriatic patients’ groups there was a statistically significant correlation between WHR and psoriasis duration noticed, respectively for Group 1 (r = 0.546, *p* = 0.03) and Group 2 (r = 0.60, *p* = 0.04) (Figure 3). However, we did not observe any statistically significant correlation between BMI and psoriasis duration, PASI, or BSA as well as between WHR and PASI, or BSA.

We observed decreased levels of miR-22, miR-133a, miR-146a, and miR-369 in both psoriatic patient groups in comparison to the control group and the differences were statistically significant (Figure 4a–d). There was also a statistically significant difference between Group 1 and Group 2 for miR-22 and miR-146a levels (Figure 4a,c). The difference between Group 1 and Group 2 for miR-133a and miR-369 was not statistically significant. There were not any statistically significant differences between groups for the Let-7b level (Figure 4e).

In psoriatic patients, in both groups as well as in the control group we have performed the correlations between selected miRNA levels and age, BMI, WHR, lipid disturbances, hypertension, hyperglycemia, smoking addiction, and alcohol abuse. Additionally, in the psoriatic patients, we performed the correlation between selected miRNA levels and psoriasis duration, PASI, and BSA. We have noticed statistically significant correlations between miR-22 and BSA, BMI; miR-146a and BMI, Let-7b and PASI, BSA. We did not find any other correlations.

We observed a positive correlation between PsA and miR-22 as well as miR-146a levels in Group 2, respectively for miR-22 r = 0.717, *p* = 0.03 (Figure 5a,b) and for miR-146a r = 0.666, *p* = 0.03 (Figure 6a,b). There was a positive correlation between miR-133a level and PsA in Group 1 (r = 0.560, *p* = 0.03) (Figure 7a,b).

The assessment of the severity of psoriasis in relation to the miRNA level showed a positive correlation between miR-22 level and BSA in Group 1 (r = 0.564, *p* = 0.025) (Figure 5c,d). There was also a positive correlation between Let-7b level and PASI as well as Let-7b level BSA in Group 1, respectively, r = 0.532, *p* = 0.04 and r = 0.569, *p* = 0.02 (Appendix A). We did not observe any statistically significant correlation between miR-22 level and psoriasis duration or PASI. There was not any statistically significant correlation between miR-133a level, miR-146a, or miR-36 and psoriasis duration, PASI, or BSA.

The assessment of obesity/overweight measurements in relation to miRNAs showed a negative correlation in Group 2 between miR-22 level and BMI (r = −0.6, *p* = 0.04) (Figure 5e,f), as well as between miR-146a level and BMI (r = −0.669, *p* = 0.02) (Figure 6c,d). We also did not observe any statistically significant correlation between all assessed miRNAs and WHR, as well as miR-369, miR-133, or Let-7b levels and BMI. In the control group, we also did not observe any correlations between BMI or WHR and selected miRNA.

## 3. Discussion

There are observations that overweight and obesity can be associated with psoriasis; furthermore, it can be a risk factor for psoriasis onset and often a worse response to treatment among psoriatic patients [23,24,25,26,27,28,29,30,31]. It was found that the prevalence of psoriasis is increasing in people with high body mass index (BMI) [24,27,30,31] and the psoriasis severity is correlated with increased BMI [23,26]. A few links have been indicated between psoriasis and obesity, including gut dysbiosis, intestinal inflammatory response to a high-fat diet, proinflammatory environment within adipose tissue, impairment in intestinal permeability leading to the passage of pro-inflammatory toxic substances and immune system modulators into the circulatory stream [26,30]. For the first time, we detected a statistically significant difference in the levels of miR-22 and miR-146a between psoriatic patients with overweight or obesity (Group 1) and psoriatic patients with normal range of weight (Group 2). This may suggest that these mi-RNAs may have an impact on the obesity-CVD-psoriasis network.

According to the WHO, obesity is defined as BMI ≥ 30 kg/m^2^ and overweight is classified as BMI 25.0 to <30 kg/m^2^, normal weight ranges from BMI of 18.5 to <25 kg/m^2^ [43]. BMI is a common measure to diagnose obesity; however, it has some limitations. The measure itself cannot distinguish between fat mass and lean muscle mass [44,45]. Additional measurements like waist circumference or WHR can be helpful to assess body fat distribution and they are more strongly correlated with visceral adipose tissue than BMI and are superior to BMI in predicting CVD risk [46]. According to the WHO, there are cutoffs of 102 cm in men and 88 cm in women for waist circumference, or of ≥0.90 in men and ≥0.85 in women for WHR [44,46]. We have used two types of measurements—BMI and WHR—to define obesity and overweight in the psoriatic patients’ groups and in the control group for better group characteristics. There were statistically significant differences in BMI and WHR between Group 1 and Group 2, and the control group, with no statistically significant differences between Group 2 and the control group. However, we did not observe any statistically significant correlation between BMI and psoriasis duration, PASI or BSA. It is consistent with the results of our previous studies [47]. Some authors noticed that the psoriasis severity is correlated with increased BMI [23,26]. We have found limitations of separate BMI measurements, in view of this we added WHR measurement for this study. We have detected a statistically significant correlation between WHR and psoriasis duration for Group 1 and Group 2. The chronic inflammation and increased production of IL-6 and IL-17A in psoriasis influence adipocytes differentiation and their capability of adipokine and chemokine synthesis and increase in visceral and subcutaneous white adipose tissue (WAT) [35]. Consequently, WHR measurement can be more significant than BMI in the correlations with psoriasis characteristics.

We have observed decreased levels of miR-22, miR-133, miR-146, and miR-369 in psoriatic patients. For miR-22 and miR-146 there were also statistically significant differences noted between obese/overweight patients and those with normal weight. miR-22 is associated with psoriasis and CVD; however, there are not any additional associations with overweight or obesity so far. miR-146a dependence with CVD, psoriasis, and obesity are noted in the constructed miRNA-disease network (Figure 1).

The decreased miR-22 levels were reported in the blood of patients affected by psoriasis vulgaris compared to healthy subjects by other authors [20,48]. Alatas et al. suggested that miR-22 may be inhibited by signal transducer and activator of transcription 3 (STAT3) which is hyperactivated in psoriasis [20]. STAT3 mediates the signal of most cytokines that are involved in psoriasis pathogenesis, including the IL-23/IL-17/IL-22 axis. The accumulation of adipose tissue in overweight or obese patients induces low-grade inflammation and extended pro-inflammatory cytokine production, and hence, the activation of STAT3 signaling [49]. As a consequence, the miR-22 level in PBMCs of Group 1 was intensively decreased. These observations may also explain the higher risk of CVD in psoriatic patients, especially in overweight or obese ones. A decreased miR-22 level was associated with heart failure (HF) worsening [50,51,52] and it was observed in arteries with atherosclerotic plaques compared to healthy arteries [53,54] and in patients affected by acute myocardial infarction (AMI) compared to the controls [55]. On the other hand, increased miR-22 levels were observed in cardiomyocyte hypertrophy and in different cardiac hypertrophy mice models [56,57]. Gurha et al. showed that miR-22 overexpression induced cardiac dilation and HF in mice [58]. Gupta et al. found that pharmacological inhibition of miR-22 after heart infarction improved cardiac function and inhibited cardiac remodeling in older mice [59]. They also demonstrated an association between higher baseline levels of circulating miR-22 and mortality in 154 HF patients. 

The role of miRNA-146a in the pathogenesis of psoriasis is not fully revealed and previously published data concerning its level are ambiguous. Xia et al. [60] and Garcia–Rodriguez et al. [61] observed elevated levels of miR-146a in PBMCs of psoriasis patients. After MTX or topical treatment, miRNA-146a expression was even higher than before treatment and in the control group [61]. Overexpression of miR-146a was also found in lesional skin [15,16,60,62,63]. Garcia-Rodriguez et al. observed elevated levels of miR-146a in psoriatic patients’ plasma and its decrease after treatment [61]. Leal et al. have found higher miR-146a serum levels in patients with psoriasis than in controls [22]. However, decreased plasma or serum miR-146a levels were also noticed in psoriatic patients [64,65]. miR-146a-deficient mice showed psoriasis-like lesions with skin thickness, keratinocyte hyperproliferation, production of chemokines, and leukocyte skin infiltration. In vivo delivery of miR-146a reduced these disturbances in mice [66]. miRNAs are extremely pleiotropic molecules, which can regulate and be regulated by multiple mechanisms and these results suggest that miR-146a activities can differ depending on the type of tissue and stimuli [61]. Previous studies indicated that miR-146a may act as a protective factor or a disease promoter in psoriasis [67]. Increased levels of miR-146a can target TNF receptor-associated factor 6 (TRAF6) and IL-1 receptor-associated kinase 1 (IRAK1) and cause the inhibition of IL-6, IL-8, TNF-α, and C-C Motif Chemokine Ligand 20 (CCL20) expression, and consequently lead to suppression of neutrophil chemoattraction by keratinocytes [62,67,68]. On the other hand, miR-146a can also induce the activity of inflammatory chemokines, via downregulating atypical chemokine receptor 2 (ACKR2) in keratinocytes and increase T cell infiltration in psoriatic epidermis [67,69]. The miR-146a immunomodulatory role can be also explained by the observation that miR-146a is preferentially expressed by Treg cells [15,70]. In our study, a statistically significant decreased level of miR-146a in PBMCs, especially in the patients with overweight or obesity, indicated that miR-146a may be the protective factor in psoriasis. A decreased level of miR-146a can lead to the chronic inflammation observed in psoriasis. In obese/overweight psoriatic patients the additional influence of proinflammatory cytokines produced by the adipose tissue may cause exacerbation of this state. These results may be also supported by Olivieri et al.’s observation that in the presence of proinflammatory stimuli, IL-6 levels increase and miR-146a levels decrease [71].

In relation to miR-146a, similar to psoriasis, there are also some uncertainties observed in CVD [48,72]. The increased miR-146a expression was reported in atherosclerotic plaques, leading to disease progression [72,73,74,75]. However, some studies indicated that miR-146a might reduce the oxidized low-density lipoprotein (oxLDL)-induced inflammatory response [76,77,78]. miR-146a was found to be upregulated in human hearts in HF patients [79], as well as in plasma and PBMCs of AMI patients [80,81]. It was found that the Th1/Th2 imbalance has an important role in AMI pathogenesis and the overexpression of miR-146a is followed by changing the response toward Th1 inflammation, although miR-146a is recognized typically as an inhibitor of Th1 cells responses [72,82]. These observations in CVD confirm that many factors can be involved in the miR-146a aberrant regulatory function and increase in the Th1 cells the same as in psoriasis. It will be valuable to find out which factors can influence the individual immunity switch.

Epigenetic processes affecting gene expression without changes in the nucleotide sequence may contribute to the pathophysiology of inflammatory diseases such as psoriasis, CVD, and obesity. It has been documented that epigenetic modifications occur after environmental stimuli and play a very important role in inflammatory gene transcription [11,12,13,83,84]. Diet, obesity, air pollution, chemical exposure, some medications, bacterial/viral infections, smoking, excessive alcohol consumption, sleep deprivation, chronic stress, and low physical activity could have effects on epigenetic modifications and trigger susceptibility to diseases [11,12,13,83,84]. However, elucidation of the specific epigenetic pathways involved in the modulation of inflammatory and anti-inflammatory genes is still unknown. In psoriatic patients, in both groups as well as in the control group, we have performed the correlations between selected miRNA levels and age, BMI, WHR, lipid disturbances, hypertension, hyperglycemia, smoking addiction, and alcohol abuse. Additionally, in the psoriatic patients, we performed the correlation between selected miRNA levels and psoriasis duration, PASI, and BSA. We have noticed statistically significant correlations between miR-22 and BSA, BMI; miR-146a and BMI, Let-7b and PASI, BSA. We did not find any other correlations.

The negative correlation between miR-22 level and BMI as well as miR-146 and BMI only in psoriatic patients with normal weight may suggest that these mi-RNAs may be involved in the obesity–CVD–psoriasis network. According to other authors, miR-146a was also inversely related to BMI [85]. Downregulation of miR-146a was observed in overweight and obese adolescents [86], as well as in Algerian obese men and miR-146a level negatively correlated with IL-6, TNF-α, and CD36 [87]. In the obese/overweight psoriatic patients, the severity of the disease was more intensive according to the BSA and PASI scores. The more intensive inflammatory process may be the result of additional stimulation of STAT3 signaling [49]. In the group with normal weight, the impact of extensive adipose tissue is reduced, therefore the association of miR-22 or miR-146 with BMI may be noticed more easily. 

In the assessment of disease severity, we noticed a positive correlation between miR-22 level and BSA, as well as Let-7b level and PASI, and BSA in Group 1. We have expected a negative correlation between miR-22 or Let-7b and BSA or PASI; however, the results may also suggest that other regulatory factors could have effects on epigenetic modifications in these interactions.

In the constructed miRNA-disease network, only one dependence of miR-133a with CVD is noted (Figure 1). miR-133a was found to be an inhibitor of brown adipose tissue (BAT) differentiation [38,88,89,90]. BAT activity increases the body’s energy expenditure, for that reason, it may have a protective function against obesity and its complications [38,91,92,93]. BAT activity is impaired in obese people and correlates inversely to BMI, body fat content, and central adiposity [38,94,95]. However, in our psoriatic patients, we observed decreased levels of 133a, and the difference between Group 1 and Group 2 was not statistically significant. Decreased miR-133a level in psoriasis can explain the onset of CVD in these patients, because in CVD miR-133a level is also downregulated, like in the heart muscle in the area of infarction and in the border zone in humans and in experimental animals [48,96,97]. Decreased level of miR-133a is involved in cardiac hypertrophy [98]. Downregulation of miR-133a in plasma, in atherosclerotic plaques, and in vascular smooth muscle cells (VSMCs) was also found [99]. A decrease in miR-133a inhibits the osteogenic differentiation of VSMCs and induces arterial calcification [100]. On the contrary, serum miR-133a level is increased in AMI patients or with unstable angina pectoris, and there was a positive correlation between serum miR-133a level and the infarcted area [101].

Most authors observed elevated levels of miR-369 in psoriatic patients’ serum and skin in comparison to healthy controls [1,16,20,21,102]. In some studies, correlations of serum and skin miR-369 levels with disease severity were confirmed [21,102]. Despite reports of increased levels of miR-369 in psoriatic patients’ serum and skin, our study shows for the first time that miR-369 levels in PBMCs are downregulated in psoriatic patients. This can be explained by the observation that miR-369 decreases lipopolysaccharide (LPS)-induced nitric oxide (NO) production by targeting inducible nitric oxide synthase (iNOS) expression and inhibiting nuclear translocation of transcription factors NF-κB [48,103]. Therefore, it is regarded as a key regulator of inflammatory response in dendritic cells. miR-369 overexpression in these cells significantly decreased proinflammatory cytokine production like TNF-α, IL-6, IL-12, IL-1α, and IL-1β in response to LPS [48,103]. Most PBMCs are lymphocytes (70–90%) and Th cells contribute to the most significant portion of lymphocytes (45–70%). The pathology of psoriasis is strongly associated with Th1, Th17, and Th22 response so patients with severe psoriasis without any systemic treatment may present a decreased level of miR-369 in PBMCs and high level of proinflammatory cytokines. The results in CVDs of miR-369 presented both decreased and increased values. miR-369 was downregulated in myocardial tissue after AMI in rats [104]. The plasma of patients with HF, following AMI, exhibited higher miR-369 expression levels, compared to subjects affected by AMI without HF [105]. More studies are needed to explain these discrepancies.

There were not any statistically significant differences between groups for the Let-7b level. According to most of the studies, the Let-7b level was decreased in psoriatic skin [1,48,106,107]. There were also decreased levels of Let-7b in plasma-derived extracellular vesicles noticed in PsA compared to cutaneous psoriasis only [108]. Let-7b was indicated as a potential candidate regulator in acute myocardial infarction by weighted gene co-expression network analysis [109]. Let-7b was downregulated in human carotid plaque tissues in symptomatic compared to asymptomatic atherosclerotic patients. Decreased levels of Let-7b were detected in diabetic plaque tissues compared to non-diabetic tissues, which can indicate that diabetes may be an accelerator of the atherosclerosis process [48,110].

We have noticed a positive correlation between miR-133a level and PsA in all psoriatic patients and there was also a positive correlation between miR-133a level and PsA in Group 1. That can indicate that miR-133a can be also a marker of PsA and overweight or obesity can be an additional risk factor for PsA. miR-133a level was found to be decreased in the lesional versus non-lesional skin of psoriatic patients and its level increased after treatment with biologics [48]. Plasma levels of miR-22 were found to be markedly lower in PsA subjects [48,111]. On the other hand, miR-146a was found to be overexpressed in PsA compared to healthy controls with higher levels observed in patients with higher disease activity [112]. Interestingly, decreased levels of miR-146a were observed in PsA with a poor therapeutic response [112]. We also noticed a positive correlation between miR-22, miR-146a and PsA but only in the group of patients with normal weight. Previously, it was found that obese PsA patients present a different clinical response to systemic treatment, especially to TNF-α blockers [113,114]. It can be explained by the influence of fat mass on drug distribution [113,114]. There are also observations that the inhibition of T cell co-stimulation is not influenced in obese patients [115]. The inadequate response to TNF-α blockers can be also dependent on elevated circulating levels of Th17 cells and IL-17 in PsA [116,117]. Potentially, overweight or obesity can also influence epigenetic modulation in PsA; therefore, miR-22 and miR-146 correlations differ in these two groups of patients. Unfortunately, the current understanding of psoriatic arthritis (PsA)-associated epigenetic marks is limited and detailed investigations are required to understand their role in molecular pathophysiology. The molecular mechanisms of miRNA-146a are best understood [118]. miRNA-146a binds the mRNA 3′-UTR of TNF receptor-associated factor 6 (TRAF6) and the IL-1R-associated kinase (IRAK1), inhibiting their expression [72,119]. The pathogen-associated molecular patterns (PAMP) and damage-associated molecular patterns (DAMP) direct Toll-like receptors (TLRs) and IL-1 receptors that engage the myeloid differentiation primary response 88 (MyD88) and IRAK family proteins. The activation of TRAF6-associated IRAK proteins induces osteoclast differentiation and inflammation [120]. In PsA progressive bone destruction is mediated by monocyte-derived osteoclasts. Lin et al. assessed miR-146a in CD14⁺ monocytes and monocyte-derived osteoclasts to find out if it causes active osteoclastogenesis in PsA patients. The results showed that the miR-146a-5p expression was higher in PsA patient-derived CD14⁺ monocytes compared to patients without PsA and the control group. After clinical improvement using biologics, the miR-146a expression in CD14⁺ monocytes from PsA patients was selectively reduced. These results indicate that miR-146a expression in CD14⁺ monocytes derived from PsA patients correlates with clinical efficacy and the induction of osteoclast activation and bone resorption [121]. Chatzikyriakidou et al. observed that *IRAK1* rs3027898 polymorphism can be associated with PsA susceptibility [119]. Caputo et al. found that in genetic variants affecting miRNA-146a (rs2910164 c.60C > G, MIR146A), the presence of the G allele has been associated with higher susceptibility to PsA [122]. It seems also that miRNAs may be the marker of drug response in psoriasis.

## 4. Materials and Methods

### 4.1. Studied Groups

The study was conducted on patients from the Outpatient Clinic of the Department of Dermatology, Venereology and Pediatric Dermatology, Medical University of Lublin, Poland. The study comprised 28 male psoriatic patients with severe psoriasis, who have suffered from psoriasis for longer than 5 years, with no history of systemic treatment for at least 4 weeks prior to entering the study and 16 male healthy controls. We also preferred the psoriatic patients with hypertension (blood pressure 140/90 mmHg or higher) and/or lipid disturbances (hypertriglyceridemia and/or hypercholesterolemia) to assess their cardiovascular disease risk. The psoriatic patients were divided into two groups by BMI and waist–hip ratio (WHR). We selected patients with BMI ≥ 25 kg/m^2^ and WHR ≥ 0.9 to Group 1 and patients with BMI < 25 kg/m^2^ and WHR < 0.9 to Group 2. The characteristics of both psoriatic groups and the control group are presented in Table 1.

In psoriatic patients, in both groups as well as in the control group we have performed the correlations between selected miRNA levels and age, BMI, WHR, lipid disturbances, hypertension, hyperglycemia, smoking addiction, and alcohol abuse. Additionally in the psoriatic patients we performed the correlation between selected miRNA levels and psoriasis duration, PASI (Psoriasis Area and Severity Index) and BSA (Body Surface Area). 

### 4.2. Assessment of Psoriasis Severity

The skin lesion severity was assessed with the use of PASI (psoriasis area and severity index) and BSA (body surface area). Psoriasis arthritis (PsA) was recognized according to CASPAR criteria [123].

### 4.3. miRNA Isolation

We have chosen peripheral blood mononuclear cells (PBMCs), because the pathology of psoriasis as well as cardiovascular disease is strongly associated with Th1, Th17, and Th22 response. Most PBMCs are Th cells. Therefore, miRNA isolated from PBMCs seems to be the best material for the assessment. PBMCs were isolated by density gradient centrifugation using Gradisol L (Aqua-Med, Łódź, Poland). miRNA was isolated from PBMCs using the Syngen miRNA Kit (Syngen, Wrocław, Poland) according to the manufacturer’s protocol. The concentration and purity of miRNAs were evaluated by NanoDrop 2000 (ThermoFisher Scientific, Waltham, MA, USA). The samples were stored at −80 °C.

### 4.4. Quantitative RT-PCR (RT-qPCR)

Extracted miRNA was reverse transcribed using the TaqMan MicroRNA Reverse Transcription Kit (ThermoFisher Scientific, Waltham, MA, USA). RT-qPCR was performed using TaqMan Universal PCR Master Mix (ThermoFisher Scientific, Waltham, MA, USA) and TaqMan miRNA Assays (ThermoFisher Scientific, Waltham, MA, USA) following the manufacturer’s protocol. The specific TaqMan miRNA Assays were used in the experiments: miR-22-3p (hsa-miR-22, ID 000398), miR-133a-3p (hsa-miR-133a, ID 002246), miR-146a-5p (hsa-miR-146a, ID 000468), miR-369-3p (hsa-miR-369-3p, ID 000557), Let-7b-5p (hsa-let-7b, ID 002619), RNU48 (RNU48, ID 001006). RT-qPCR was performed by QuantStudio 3 (ThermoFisher Scientic, Applied Biosystems, Foster City, CA, USA). The data were normalized relative to RNU48. All the PCR reactions were run in triplicates. The relative expression was calculated by the formula RQ = 2^−ΔΔCt^ [124].

### 4.5. The Constructed miRNA-Disease Network

We performed the analysis of the associations between selected miRNA and CVD, obesity or psoriasis with miRNet: https://www.mirnet.ca/miRNet/home.xhtml, accessed on 14 May 2023 (Figure 1) [42]. miRNet 2.0 is a platform designed to help elucidate miRNA functions by integrating users’ data with existing knowledge via network-based visual analytics. In our analysis, we have used the list of selected miRNAs: hsa-miR-22, hsa-miR-133a, hsa-miR-146a, hsa-miR-369-3p, and hsa-let-7b in Homo sapiens (human) with miRBase ID and we have chosen disease as targets. Then, the input lists were mapped to the underlying knowledge bases to create networks. miRNet 2.0 can automatically recognize different versions of miRBase IDs. The miRNA interaction knowledgebase is based on microRNA database (miRBase), microRNA-target interactions database (miRTarBase), TarBase, the human microRNA disease database (HMDD), tissue-specific miRNAs (TSmiR), and an interactive multi-omics-tissue atlas (IMOTA).

### 4.6. Data Analysis

The data were plotted as the mean value ± standard error of the mean (SEM) and analyzed using GraphPad Prism 8 software (GraphPad Software, Inc., La Jolla, CA, USA). Statistical analysis was performed using Mann–Whitney’s U test or Spearman’s correlation (significance was accepted at *p* < 0.05).

## 5. Conclusions

miRNA seems to be an important aspect of studies in psoriasis nowadays, which may help us better understand different mechanisms in psoriasis pathogenesis and, thus, implement new therapeutic methods. Many studies have been performed so far; however, the lack of correlation between the various tissues suggests an incomplete understanding of the regulatory mechanisms of miRNA expression in skin, plasma, serum, and PBMCs. The lack of endogenous control groups in some studies can have also an impact on the differences in results. Standardized study designs, isolation techniques, miRNA expression profiling platforms, and data analysis are needed for better result interpretations.

We have found decreased levels of miR-22, miR-133a, miR-146a, and miR-369 among psoriatic patients. The decreased levels of selected miRNA are consistent with the levels observed in CVD indicating their impact on the CVD risk in psoriatic patients. There was a statistically significant difference in miR-22 and miR-146a levels between the patients with obesity or overweight and the patients with normal weight. These results may suggest that miR-22 and miR-146 may be recognized a contributing factor in the obesity-CVD-psoriasis network. However, the study should be extended to more patients and the control group (without psoriasis) with a similar BMI and WHR to G1 could be added for more significant results. The molecular mechanisms of these relationships have not been fully elucidated so far. Detailed genetic, cellular, and in vivo investigations are required to understand the role of miR-22 and miR-146a in the pathogenesis of obesity, CVD and psoriasis network. We are planning the study with miR-22 and miR-146a mimics and inhibitors on immune cells, as well as adipocytes and endothelial cells which are engaged in CVD. miRNA can mediate obesity-induced endothelial dysfunction by affecting gene expression of endothelial nitric oxide synthase (eNOS), Sirtuin 1 (SIRT1), oxidative stress, autophagy machinery, and endoplasmic reticulum (ER) stress [125]. It will be very important to recognize the role of miR-22 and miR-146a in these processes. The identification of miR-22 and/or miR-146a as new targets for interventions may prevent or delay the development of obesity-related cardiovascular disease in psoriatic patients. In vivo the assessment of the phenotype of a knock out/knock down of selected miRNAs in models of psoriasis could be performed. This way of study design has already been proved in the assessment of the role of T cells or cytokine networks in psoriasis when the initial observations in psoriatic patients were confirmed, validated, and extended by animal models of psoriasis [126,127,128].

Potentially, overweight or obesity can also influence epigenetic modulation in PsA, therefore we observed positive correlations between miR-22 and miR-146a levels and PsA in Group 2 and between miR-133a level and PsA in Group 1. It can be also related to some gene polymorphism or other environmental factors. This part of the study should be also continued.

The study could be also extended to a transcriptomic analysis of PBMCs for better knowledge of other miRNAs that correlate with the high risk of CVD.

## Figures and Tables

**Figure 1 ijms-24-13916-f001:**
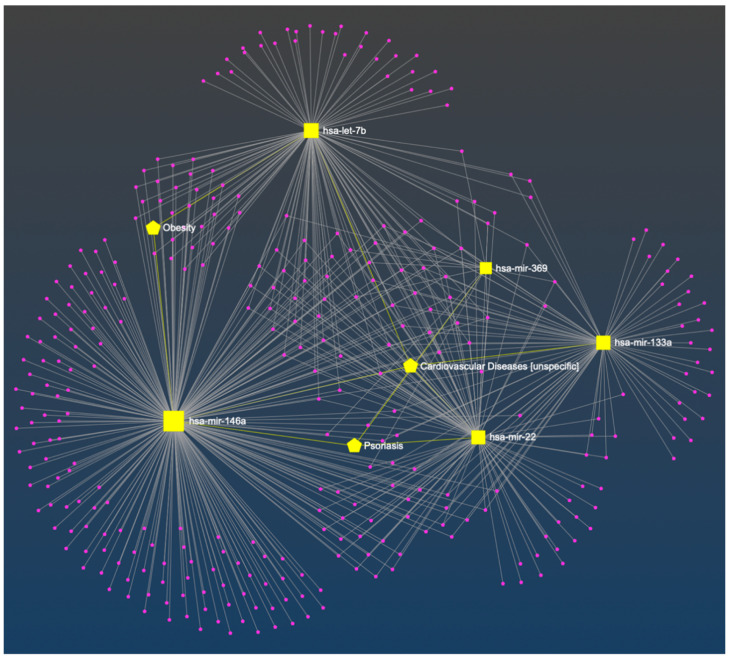
The constructed miRNA-disease network. The bipartite network was composed of miRNAs (squares) and diseases (pentagon). The associations between selected miRNA and cardiovascular disease, obesity, or psoriasis are marked in yellow [42].

**Figure 2 ijms-24-13916-f002:**
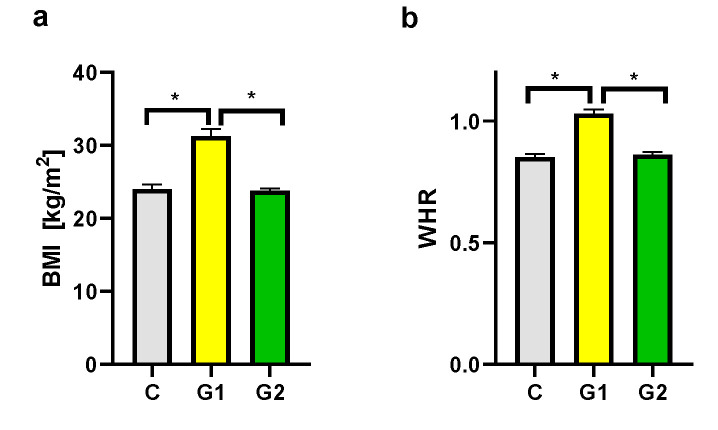
BMI (**a**) and WHR (**b**) in groups. C—control group, G1—Group 1: psoriatic patients with obesity or overweight, G2—Group 2: psoriatic patients with normal weight. Data represent the mean value ± SEM. Values significant (*) with *p* < 0.05 (Mann–Whitney’s U test).

**Figure 3 ijms-24-13916-f003:**
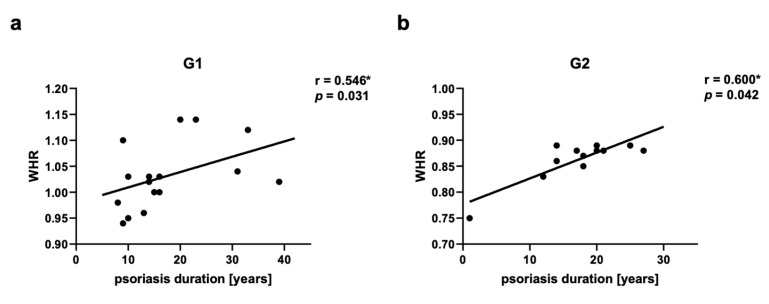
Correlations between waist-hip ratio (WHR) and psoriasis duration in psoriatic patients’ groups G1 (**a**) and G2 (**b**). G1—Group 1: psoriatic patients with obesity or overweight, G2—Group 2: psoriatic patients with normal weight. Values significant (*) with *p* < 0.05 (Spearman’s correlation).

**Figure 4 ijms-24-13916-f004:**
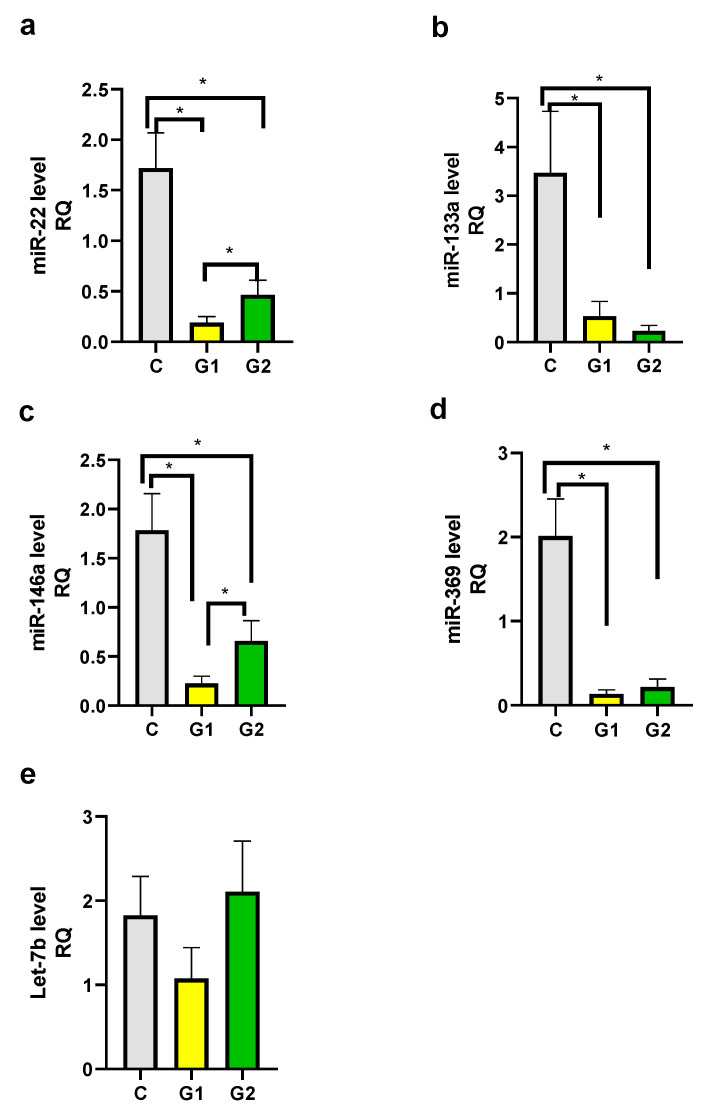
mi-RNA level in groups. (**a**) miR-22, (**b**) miR-133a, (**c**) miR-146a, (**d**) miR-369, (**e**) Let-7b. C—control group, G1—Group 1: psoriatic patients with obesity or overweight, G2—Group 2: psoriatic patients with normal weight. Data represent the mean value ± SEM. Values significant (*) with *p* < 0.05 (Mann–Whitney’s U test).

**Figure 5 ijms-24-13916-f005:**
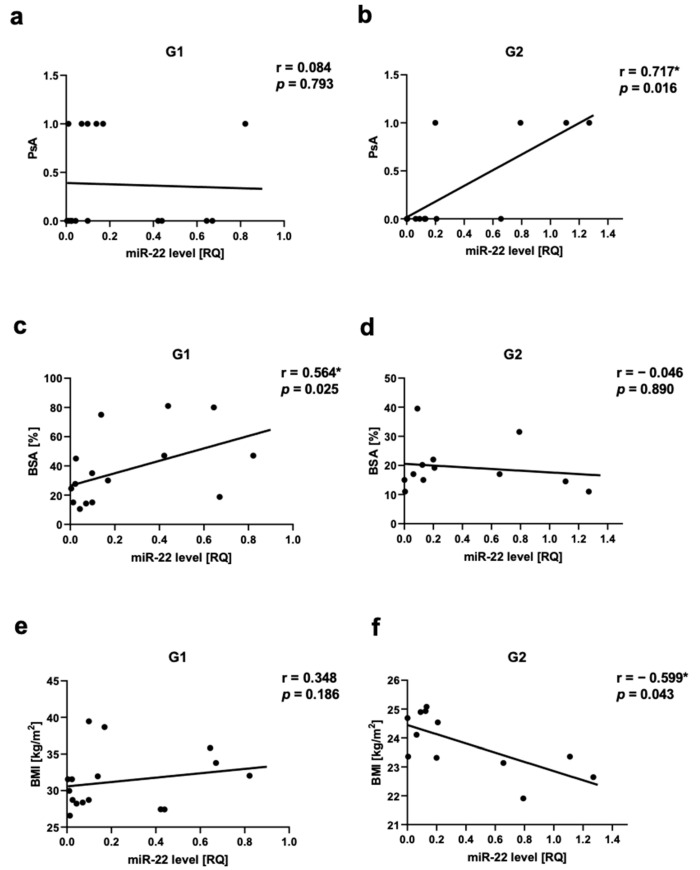
miR-22 correlations in psoriatic patients’ groups. (**a**,**b**)—with psoriatic arthritis (PsA), (**c**,**d**)—with body surface area (BSA), (**e**,**f**)—with body mass index (BMI). G1—Group 1: psoriatic patients with obesity or overweight, G2—Group 2: psoriatic patients with normal weight. Values significant (*) with *p* < 0.05 (Spearman’s correlation).

**Figure 6 ijms-24-13916-f006:**
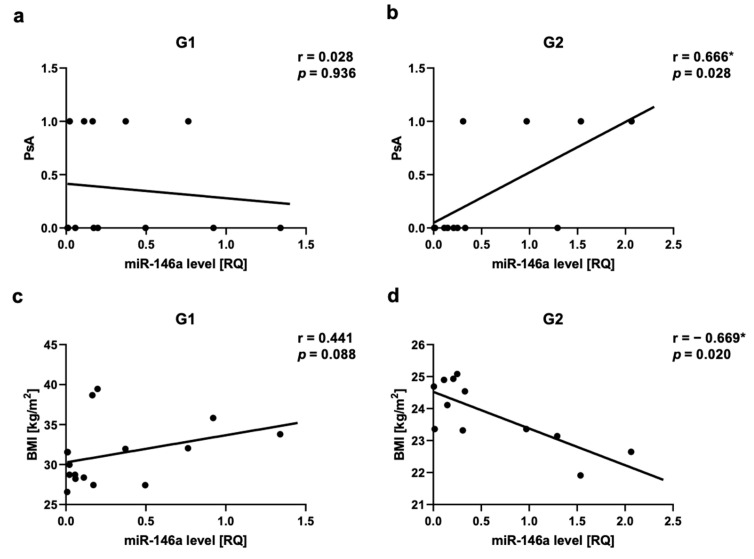
miR-146a correlations in psoriatic patients’ groups. (**a**,**b**)—with psoriatic arthritis (PsA), (**c**,**d**)—with body mass index (BMI). G1—Group 1: psoriatic patients with obesity or overweight, G2—Group 2: psoriatic patients with normal weight. Values significant (*) with *p* < 0.05 (Spearman’s correlation).

**Figure 7 ijms-24-13916-f007:**
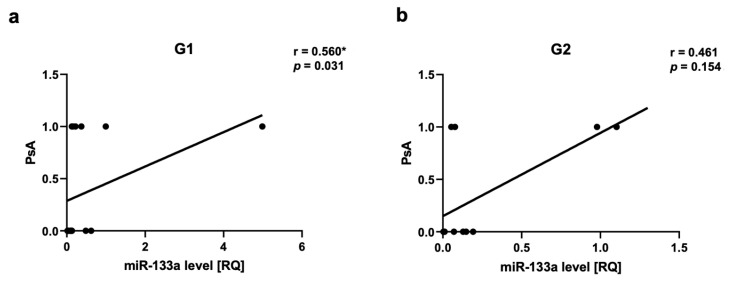
miR-133a correlations in psoriatic patients’ groups. (**a**,**b**)—with psoriatic arthritis (PsA). G1—Group 1: psoriatic patients with obesity or overweight, G2—Group 2: psoriatic patients with normal weight. Values significant (*) with *p* < 0.05 (Spearman’s correlation).

**Table 1 ijms-24-13916-t001:** Demographic and clinical characteristics of patients and control group.

Characteristic	Psoriatic Patients		Control Group (*n* = 16) Mean (SEM)
	Group 1 (*n* = 16) Mean (SEM)	Group 2 (*n* = 12) Mean (SEM)	
Age (years)	45.9 (13.5)	40 (12.3)	38.4 (8.3)
Duration of psoriasis (years)	17.5 (9.2)	17.25 (6.7)	-
Psoriatic arthritis (PsA) *	6 (37.5%)	4 (33.3%)	-
PASI	21.7 (9.7)	16.35 (5.98)	-
BSA	36.3 (24.2)	19.4 (8.4)	-
BMI	31.3 (3.9)	23.8 (1.0)	24.0 (2.4)
BMI in patients with PsA	31.6 (3.8)	22.8 (0.7)	24.0 (2.4)
WHR	1.03 (0.1)	0.86 (0.04)	0.85 (0.05)
WHR in patients with PsA	1.08 (0.06)	0.88 (0.01)	0.85 (0.05)
Lipid disturbances *	10 (62.5%)	6 (50%)	3 (18.75%)
Hypertension *	9 (56.25%)	6 (50%)	2 (12.5%)
Hyperglycemia *	3 (18.75%)	1 (8.3%)	1 (6.25%)
Smoking addiction *	6 (37.5%)	8 (66.7%)	5 (31.25%)
Alcohol abuse *	2 (12.5%)	4 (33.3%)	0

* number (%).

## Data Availability

The data presented in this study are available on request from the corresponding author.

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
