# Peer review of "Selected miRNA and Psoriasis—Cardiovascular Disease (CVD)—Overweight/Obesity Network—A Pilot Study"

_ijms, 2023, doi:10.3390/ijms241813916_

Round 1

Reviewer 1 Report

In this study, the authors investigated selected miRNA expression patterns in psoriasis patients with and without obesity. The manuscript's main readouts are network analysis, qRT-PCR, and co-relation analysis. The investigated question is unique. This will shed light on miRNA-mediated regulation of psoriasis-associated co-morbidities.   

1.     Explain the disease-miRNA network analysis in detail. It needs to be clarified whether the observations in Figure 1 are predictions or based on validated research.

2.     Why were PBMCs selected over the skin, adipose tissue, plasma, or serum?

3.     A control group (without psoriasis) with a similar BMI and WHR to G1 should be added to compare the expression of selected miRNAs. This will confirm if the observed changes are common for obesity and overweight or specific to psoriasis-associated obesity and overweight.

4.     As the Let-7b expression was not significantly altered, figure 8 can be moved to supplemental figures.

5.     Concise and streamline discussion and conclusion section to make it focused for the readers. 

Reviewer 2 Report

The study contributes to the growing body of research addressing the intricate interplay between psoriasis, cardiovascular disease, and obesity. By examining miRNA profiles, the authors attempt to uncover molecular insights into the mechanisms underlying these conditions. The results indicate a significant reduction in the levels of miR-22, miR-133a, miR-146a, and miR-369 among psoriatic patients, reinforcing the potential relevance of these miRNAs in disease pathogenesis.

One aspect that warrants further discussion is the sample size employed in the study. With a cohort of 28 male psoriatic patients and 16 male healthy controls, the sample size appears to be limited. Given the multifaceted nature of the investigated conditions and the inherent variability within patient populations, a more substantial sample size could enhance the statistical power and generalizability of the findings. A larger sample could provide a more comprehensive perspective on the relationship between miRNA profiles, body weight, and disease risk.

While the study demonstrates statistically significant differences in miR-22 and miR-146a levels between psoriatic patients with varying body weight statuses, the interpretation of these findings should consider potential confounding factors. Factors such as age, genetics, and medication usage could influence miRNA expression patterns. Addressing these variables could strengthen the robustness of the study's conclusions.

The identification of positive correlations between specific miRNA levels and psoriatic arthritis (PsA) within different weight categories is noteworthy. This observation suggests potential molecular markers for PsA within distinct patient subsets. However, the mechanisms underlying these correlations remain unclear. Investigating the functional roles of these miRNAs in PsA pathogenesis could provide valuable mechanistic insights.

In conclusion, the paper provides valuable insights into the connections between miRNAs, obesity, cardiovascular disease risk, and psoriasis. The findings underscore the complexity of these interactions and offer potential targets for future research and clinical applications. Nonetheless, the limitation of sample size should be acknowledged, and efforts to replicate and extend these results in larger, more diverse cohorts are essential to strengthen the validity and broader relevance of the study's outcomes.

Questions:

Could you provide further insight into the rationale behind the chosen sample size, considering the complexity of the investigated conditions and the potential variability within patient populations?

Have the authors discussed the potential impact of confounding factors, such as genetics, and medication usage, on the observed miRNA expression patterns?

Are there any plans to conduct functional studies that could provide mechanistic insights into the roles of miR-22 and miR-146a in disease progression and their potential as therapeutic targets?

Given the positive correlations between specific miRNA levels and psoriatic arthritis, are there any hypotheses or preliminary explanations for the molecular mechanisms underlying these associations?

Round 2

Reviewer 1 Report

The authors have provided adequate responses to my questions, and I do not have any further comments.